# Introducing New Cropping Pattern to Increase Cropping Intensity in Hill Tract Area in Bangladesh

Rigyan Gupta [1], Mohammad Joyel Sarkar [2], Md. Shafiqul Islam [3,*], Md. Romij Uddin [3], Israt Jahan Riza [3], Sirajam Monira [3], Farhana Zaman [3], Ahmed Khairul Hasan [3], A. K. M. Mominul Islam [3], Abeer Hashem [4], Graciela Dolores Avila-Quezada [5], Javid A. Parray [6], Elsayed Fathi Abd_Allah [7] and Uttam Kumer Sarker [3,*]

[1] Plant Breeding Division, Bangladesh Institute of Nuclear Agriculture, Mymensingh 2202, Bangladesh; rigyan2008@gmail.com

[2] Horticulture Division, Bangladesh Institute of Nuclear Agriculture, Mymensingh 2202, Bangladesh; sarkarjewel11@gmail.com

[3] Department of Agronomy, Bangladesh Agricultural University, Mymensingh 2202, Bangladesh; romijagron@bau.edu.bd (M.R.U.); isratriza7@gmail.com (I.J.R.); sirajam37297@bau.edu.bd (S.M.); fzaman@bau.edu.bd (F.Z.); akhasan@bau.edu.bd (A.K.H.); akmmominulislam@bau.edu.bd (A.K.M.M.I.)

[4] Botany and Microbiology Department, College of Science, King Saud University, P.O. Box 2460, Riyadh 11451, Saudi Arabia; habeer@ksu.edu.sa

[5] Facultad de Ciencias Agrotecnológicas, Universidad Autónoma de Chihuahua, Chihuahua 31350, Mexico; gdavila@uach.mx

[6] Department of Higher Education, Govt. Degree College, Eidgah, Srinagar 190017, India; javid06@gmail.com

[7] Plant Production Department, College of Food and Agricultural Sciences, King Saud University, P.O. Box 2460, Riyadh 11451, Saudi Arabia; eabdallah@ksu.edu.sa

* Correspondence: shafiqagron@bau.edu.bd (M.S.I.); uttam@bau.edu.bd (U.K.S.); Tel.: +880-1718330285 (M.S.I.); +880-1716809747 (U.K.S.)

**Abstract:** In Bangladesh's hill regions, where there is less cultivable land, increasing crop output requires efficient land use. Thus, in this challenging farming setting, two crop-based patterns evolved into three or four crop-based patterns. To increase cropping intensity and farmer income by incorporating mustard and mungbean in a rice-based cropping pattern, a field experiment was carried out at Sadar and Panchari Upazila, Khagrachhari during 2017–2018 and 2018–2019. Two years' mean data (using a block farming approach) showed that the modified pattern had produced a much higher yield through improved management practices. In the improved cropping pattern (Transplant *aman* (T. *aman*)–mustard–mungbean–*aus* rice), a higher rice equivalent yield (16.25 t ha$^{-1}$) was found due to the inclusion of mustard and mungbean in the existing rice-based cropping patterns T. *aman*–fallow–*boro* (9.87 t ha$^{-1}$) and T. *aman*–fallow–tomato (9.09 t ha$^{-1}$). The gross margin from the improved cropping pattern was 448,715 BDT, which was 44.26% higher than the mean gross margin (311,050 BDT) of the two existing cropping patterns. Farmers are interested in growing mustard and mungbean since both can easily cultivated in hilly areas and can yield great economic returns quickly. For the large-scale production of oil and pulse, the T. *aman*–mustard–mungbean–*aus* rice cropping pattern might be introduced in the Khagrachhari district of Bangladesh.

**Keywords:** cropping intensity; productivity; income; hill tract; Khagrachhari

## 1. Introduction

A cropping pattern specifies the timing and layout of the crops in a certain land area. A change in the proportion of land under different crops is a change in the cropping pattern of that area. The proportion of land under cultivation involving different crops at different points of time is referred to as a cropping pattern [1]. Land and water resources, two of the most important components for agricultural development, are becoming inadequate because of rapid changes in population and urbanization. As a result, in order to maximize the total benefits while adhering to a number of limitations, it is necessary to find the

most effective way to use the resources that are now available. [2]. In recent years, crop optimization has received significant attention, and various mathematical models have been developed in response to this interest [1].

Due to the increase in the population and decrease in the amount of land for the accommodation of the population, national food security is under threat. Moreover, the water resource facilities are not well developed in our country. Thus, farmers cannot produce crops during all the seasons of the year all over the cultivable areas of the country. There are three seasons in Bangladesh viz. *kharif*-1 (April to June), *kharif*-2 (July to September), and *rabi* (October to March). In the case of rice production, these are called the *aus*, *aman*, and *boro* rice seasons. In Bangladesh, during *kharif*-1 (*aus*) and *rabi* (*boro*) seasons, the farmers suffer water scarcity due to lack of rainfall. At this time, artificial water supply, i.e., irrigation using groundwater, rivers, canals, ponds, damps, and cricks, is the only way of producing crops in Bangladesh [3].

The key challenges confronting the 21st century are rising populations, limited food supplies, severe destitution, hunger, and the ruining of the environment [4]. These issues are more complicated because of the long-lasting repercussions of the COVID-19 outbreak according to the first worldwide evaluation of food insecurity and malnutrition for 2020 and some predictions of what hunger might look like by 2030. New estimates of healthy food costs and affordability relate food safety and nutrition variables to trend analysis [5]. In the coming decades, developing countries in Asia and Africa will see agricultural production expanded by about 70% [4]. Increasing agricultural productivity requires better cropping patterns and management. Thus, a new cropping pattern has been devised to maximize net profit within certain constraints to optimize resource utilization [6].

Despite focusing on rice predominantly, the cropping patterns of Bangladesh are remarkably diverse, with the other crops being chosen from a variety of non-rice cereals, pulses, oil crops, vegetables, and fiber crops. Therefore, in addition to providing rice as our principal source of dietary energy, rice-based farming systems also function as supplements to our regular meals [6,7]. Prior to 1970 viz. before the Green Revolution period, numerous cropping patterns were developed depending on the distinctive features of traditional cultivars, antiquated management techniques, socio-economic desires, and the prevailing environmental settings. However, several modifications and re-adjustments have occurred since the Green Revolution period and up to the present, leading to the development of numerous new crop cultivars and technologies capable of acclimatizing to changing surroundings. These resource-intensive innovations transformed several direct-seeded rice regions into transplanted rice regions due to their substantial input efficiency and simplified management techniques. For example, irrigation and other technological developments not only make it easier to turn *rabi* crop-producing (winter season) regions into *boro* crop-producing regions, but they also increase the land available for the cultivation of wheat, maize, and potatoes. Consequently, numerous *rabi* crops like pulses and oil seeds were unable to adapt to these systems, and their cultivation areas were reduced considerably, leading to the substantial expansion of *boro* regions [8,9].

However, out of 316 potential cropping patterns (CPs) excluding the minor ones, *boro*–fallow–T. *aman* is regarded as the most prevalent cropping pattern in Bangladesh, and accounts for 26.92% of the country's net cropped area (NCA). Additionally, the most recent CP, Barley–fallow–fallow, accounts for 0.0002% of the NCA, whereas the *boro*–fallow–fallow CP covers less than 50% of the NCA when compared to *boro*–fallow–T. *aman*. Besides, the following three CPs, fallow–fallow–T.*aman*, *boro*–*aus*–T. *aman*, and fallow–*aus*–T. *aman*, reveal that rice and non-rice cereal-containing CPs inhabit significant parts of the nation. A total of 282 CPs are employed with rice throughout Bangladesh. In contrast, ninety-two CPs in our country contain various non-rice crops due to their unique characteristics, such as in cases where vegetables are more profitable in the vicinity of a city and therefore farmers cultivate vegetables throughout the year. Similarly, in particular char lands, solely groundnut is satisfactorily farmed, giving rise to a groundnut–fallow–fallow CP. On the other hand, wheat–jute–T. *aman* is the most prominent among the 27 wheat-focused CPs,

followed by wheat–fallow–T. *aman*, while maize–fallow–T. *aman* is the most significant CP with maize and accounts for 1.18% of the NCA. Additionally, potato–*boro*–T. *aman* is the most dominating CP among 40 CPs, including major tuber crops such as potato and sweet potato followed by potato–maize–T. *aman*, while mustard–*boro*–T. *aman* is the most remarkable among 24 mustard (major consumable oil seed crop) CPs used across the nation followed by mustard–*boro*–fallow.

Grasspea, mungbean, lentil, blackgram, field pea, and felon are prominent legumes cultivated in Bangladesh during the *rabi* season, and there are 83 CPs comprising legumes, with grasspea–fallow–T. *aman* leading the pack followed by mungbean–fallow–T. *aman*. In the case of jute, there are 56 CPs that cover 9.09% of the NCA, of which the wheat–jute–T. *aman* CP is the most practiced CP followed by onion–jute–T. *aman*, whereas tobacco–jute–T. *aman* is the most common CP with tobacco followed by tobacco–maize–T. *aman*. Vegetables are cultivated on fertile, well-drained soil near cities and towns throughout the year in three seasons, two seasons, or one season on a piece of land, and the most widely used CP with vegetables is vegetables–vegetables–vegetables followed by vegetables–fallow–T. *aman* [9,10].

The existing major cropping pattern in the Khagrachhari district of the Chittagong Hill Tracts is Transplant *aman* (T. *aman*)–fallow–fallow, which covers about 75% of all cultivated area, and another, T. *aman*–fallow–*boro* rice/vegetable, that accounts for around 21% of all cultivated area [11]. In the *boro* season, farmers need more irrigation, which, given the reduction in native soil fertility for rice monoculture, is neither profitable nor appropriate for the local farmers [12]. A cropping pattern adaptation that maximizes natural resource use is needed now. About three months after harvesting T. *aman* rice, a large portion of this area's land lies fallow and then the farmers cultivate *boro* rice or vegetables like brinjal/tomato/potato. After harvesting T. *aman* rice, this hilly area can easily grow mustard and mungbean for a quick profit. *Aus* rice cultivation is more profitable than *boro* rice because it needs a shorter time to grow. By cultivating mustard, farmers can meet the oil scarcity of the country, and by cultivating mungbean, farmers can easily meet the pulse requirements of our country. Again, as a leguminous crop, mungbean also greatly boosts soil health with nutrients. Therefore, an experiment was undertaken to assess the feasibility of growing mustard and mungbean in existing rice-based cropping patterns (T. *aman*–fallow–*boro* and T. *aman*–fallow–tomato) to boost cropping intensity and production, farmer income, and job opportunities.

## 2. Materials and Methods

### 2.1. Experimental Location and Design

The experiment was executed in two Upazillas (Sadar Upazilla and Panchari Upazilla) of Khagrachhari district in three locations viz. Shalban and Thakurchara of Sadar Upazilla and North Shantipur of Panchari Upazilla in farmers' fields. The Upazillas were chosen because irrigation facilities were readily available, both to grow crops as well as to increase the intensity of cropping and productivity of the experimental areas with the induced cropping patterns T. aman–fallow–*boro* rice or T. aman–fallow–vegetables during 2017–2018 and 2018–2019. The experimental areas are located in the Northern and Eastern Hills area (AEZ-29) of Bangladesh. The geographic coordinates of the experimental site were 23.11° N longitude and 91.97° E longitude in Bangladesh. The land type was under the medium–high land category (97.0 m) and the soil in the study regions had a loamy texture with a well drainage system (Table 1). Such soils often have poor or extremely low agricultural potential for field crops but low to high potential for tree crops. Mild-to-very-steep slopes (from 15% to over 70%), a lot of monsoon rain, the medium-to-high erodibility of most soils, often low soil fertility, and more acidic soils are major drawbacks. Under grassland, the organic matter level is low (1.5%), but under forest, it is moderate (2–5%) [13,14]. In Khagrachhari (including in the three experimental sites), the dry season is warm and mostly sunny compared to the oppressively hot, harsh, and cloudy wet season. The average annual

temperature ranges between 52 °F and 92 °F, hardly dropping below 47 °F or rising above 97 °F (Figure 1).

**Table 1.** Soil type and nutrient status of the experimental area in AEZ-29 (Northern and Eastern Hills).

| Major Land Type | pH | OM (%) | Nutrient Status | | | | | |
|---|---|---|---|---|---|---|---|---|
| | | | Total N (%) | P (µg/g) | K (meq/100 g) | S (µg/g) | Zn (µg/g) | B (µg/g) |
| Highland (92%) | 3.5–7.2 | 1.71–3.4 | 0.088–0.91 | 12–18 | 0.15–0.225 | 18–27 | 0.90–1.35 | 0.30–0.45 |
| | | L–M | VL-L | L–M | L–M | L–M | L–M | L–M |

Note: L–M= Low to Medium, VL–L = Very Low to Low.

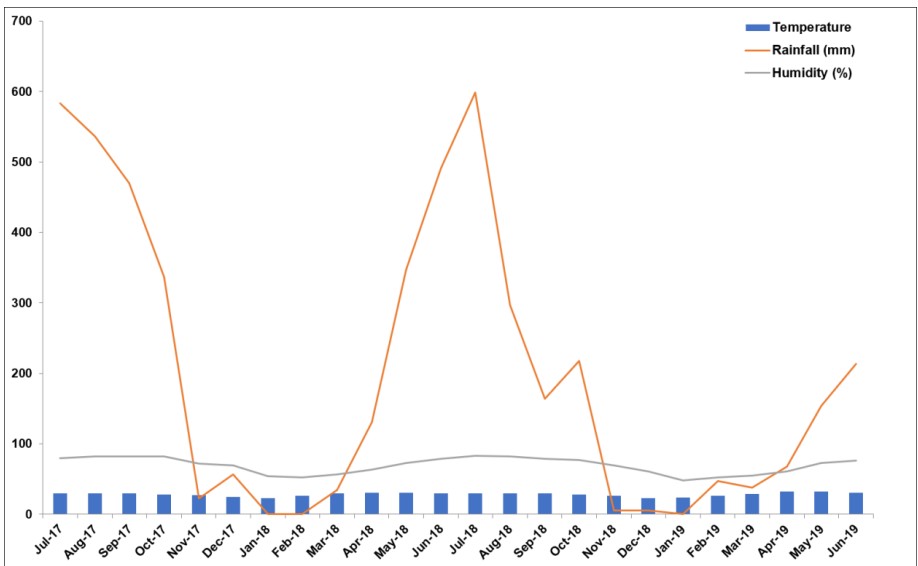

**Figure 1.** Climate data of AEZ-29 during the research period.

The experiment was laid out in a randomized complete block design with three dispersed replications (replications 1, 2, and 3 were Shalban, Thakurchara, and North Shantipur, respectively) following the block farming approach, and the unit plot size was 1335 m². Before the entire experiment was conducted again in the following year, the cropping pattern (a single factor) was replicated three times at each location in both years. For rice, field preparation, sowing technique, planting time, seedling age, weed management, pest management, rouging, etc. were calculated and done according to the cultivation procedures for the Binadhan-19 and Binadhan-17 rice variety technologies developed by the Bangladesh Institute of Nuclear Agriculture [15]. Again, for mustard and mungbean, all of the activities for the cultivation of Binasarisha-9 and Binamoog-8 were based on the variety technology developed by the Bangladesh Institute of Nuclear Agriculture [15].

*2.2. Determinations*

Data on yields and yielding characteristics, the gross income, the gross margin, the total variable cost, and the benefit cost ratio (BCR) were recorded after harvesting each crop.

Rice equivalent yield (REY): According to Verma and Modgal [16], the yield of each crop was converted into a rice equivalent using the current market values of the respective crop in order to compare the cropping patterns. The following equation was employed to calculate the REY:

$$\text{Rice equivalent yield} \left(\text{t ha}^{-1} \text{ yr}^{-1}\right) = \frac{\textit{Yield of individual crop} \times \textit{market price of that crop}}{\textit{Market price of rice}}$$

Land use efficiency (LUE): Land use efficiency was determined using the methods described by Tomer and Tiwari [17], where the total duration of each crop in a sequence is divided by 365 days. LUE is estimated using the equation mentioned below:

$$\text{Land use efficiency} = \frac{d1 + d2 + d3 + d4}{365} \times 100$$

Here, $d1$, $d2$, $d3$, and $d4$ indicate the durations of the 1st, 2nd, 3rd, and 4th crop in the sequence, respectively.

Production efficiency: According to Tomer and Tiwari [17], production efficiency was calculated as the ratio of the overall production of a cropping sequence to the entire growth period that each crop within the respective sequence needed to grow, and it was expressed in terms of kg ha$^{-1}$ day$^{-1}$. The formula used for the estimation of production efficiency is mentioned below:

$$\text{Production efficiency} = \frac{Y1 + Y2 + Y3 + Y4}{d1 + d2 + d3 + d4}$$

Here, $Y1$, $Y2$, $Y3$, and $Y4$ indicate the yields (kg) of the 1st, 2nd, 3rd, and 4th crop of the sequence, respectively. $d1$, $d2$, $d3$, and $d4$ indicate the durations (day) of the 1st, 2nd, 3rd, and 4th crop of the sequence, respectively.

Benefit cost ratio (BCR): The mean data of two cropping patterns were determined for the monetary evaluation of two separate cropping patterns. The total input cost (involving material and non-material costs) was computed to determine the expenses involved with the production of these crops. The gross return was estimated considering the current market values for the outputs of the respective crops in Bangladesh. When calculating the gross margin, the total expenditure was subtracted from the gross return, which was determined as the ratio of the gross return to the total expenditure. The benefit cost ratio (BCR) was estimated with the below-mentioned equation:

$$\text{Benefit cost ratio} = \frac{Gross\ return}{Total\ variable\ cost\ of\ production} \times 100$$

### 2.3. Data Analysis

A one-way ANOVA was carried out using the computer package Statistix 10 for Windows and significant differences among treatments were analyzed using Tukey's test at $p \leq 0.05$.

## 3. Results

### 3.1. Grain, Straw Yield, and By-Products of the Cropping Patterns

In Table 2, the grain, straw, and by-product yields, as well as the turnaround times and field durations of several cropping patterns, are displayed. According to the results, the T. *aman*–mustard–mungbean–*aus* cropping sequence was found to produce greater yields and by-products under improved farming techniques. Numerically, for the improved pattern, the average yield for T. *aman* rice was found to be 5.87 t ha$^{-1}$, for mustard it was 1.62 t ha$^{-1}$, for mungbean it was 1.46 t ha$^{-1}$, and for *aus* rice it was 5.2 t ha$^{-1}$. The average straw yields of T. *aman* rice, mustard, mungbean, and *aus* were 6.38 t ha$^{-1}$, 3.77 t ha$^{-1}$, 1.99 t ha$^{-1}$, and 5.16 t ha$^{-1}$, respectively. The total quantity of the by-products of the element crops was obtained from the mustard crop, which was about 1.02 t ha$^{-1}$, and all the data were averages for two years.

**Table 2.** Yield of different crops under existing and improved cropping patterns during 2017–2018 and 2018–2019.

| Parameters | Existing Cropping Pattern-1 | | | Existing Cropping Pattern-2 | | | Improved Cropping Pattern | | | |
|---|---|---|---|---|---|---|---|---|---|---|
| | *T. aman* | Fallow | *Boro* | *T. aman* | Fallow | Tomato | *T. aman* | Mustard | Mungbean | *Aus* |
| | Sylheti Pajam | - | BRRI dhan28 | Sylheti Pajam | - | HYV | Binadhan-17 | Binasarisha-9 | Binamoog-8 | Binadhan-19 |
| Average grain Yield (t ha$^{-1}$) | 4.52 | - | 5.35 | 4.65 | - | 9.995 | 5.87 | 1.62 | 1.46 | 5.2 |
| Average straw Yield (t ha$^{-1}$) | 5.77 | - | 6.175 | 6.13 | - | - | 6.38 | 3.77 | 1.985 | 5.16 |
| Turnaround time (days) | 80 | - | 63 | 80 | - | 60 | 12 | 10 | 8 | 12 |
| Field duration (days) | 110 | - | 112 | 110 | - | 115 | 93 | 88 | 66 | 76 |
| By-product | - | - | - | - | - | - | - | 1.02 | - | - |

### 3.2. Rice Equivalent Yield (REY)

Rice equivalent yield (REY), which was derived from the yield of the component crops, was used to compare the overall productivity of the existing cropping patterns (EP1 and EP2) and the improved cropping pattern (IP). The REY was significantly greater in the improved cropping cycle in comparison with previously used cropping cycles. The average REY of the whole improved pattern was 16.78 t ha$^{-1}$ yr$^{-1}$, where the two existing patterns' rice equivalent yields were 9.87 and 14.645 t ha$^{-1}$ yr$^{-1}$ (Table 3). The increased percentages of REY in the IP over the EPs are shown in Table 3. The REY values for EP1, EP2, and IP in 2017–2018 were 10.35, 14.02, and 16.45 t ha$^{-1}$ yr$^{-1}$, respectively, while in 2018–2019, those values were 9.39, 15.27, and 17.11 t ha$^{-1}$ yr$^{-1}$, respectively (Table 4).

**Table 3.** Mean rice equivalent yields, farmers' land use efficiencies, and production efficiencies of existing and improved cropping patterns.

| Cropping Pattern | REY (t ha$^{-1}$ yr$^{-1}$) | LUE (%) | PE (kg ha$^{-1}$ day$^{-1}$) | BCR |
|---|---|---|---|---|
| T. *aman*–fallow–*boro* | 9.87 c | 60.815 b | 41.505 b | 2.19 b |
| T. *aman*–fallow–tomato | 14.645 b | 61.635 b | 43.355 b | 2.69 a |
| T. *aman*–mustard–mungbean–*aus* | 16.78 a | 88.485 a | 50.105 a | 2.59 a |
| Increased (%) of IP over EPs | EP1 = 70.01 | EP1 = 45.50 | EP1 = 20.72 | 18.26 |
| | EP2 = 14.58 | EP2 = 43.56 | EP2 = 15.57 | 3.86 [†] |
| Level of significance | ** | ** | ** | ** |
| CV (%) | 2.20 | 1.79 | 4.77 | 4.19 |

[†] decreased (%). In a column, figures with the same letter do not differ significantly whereas figures with dissimilar letters differ significantly. ** = Significant at 1% level of probability.

### 3.3. Land Use Efficiency (LUE)

The successful exploitation of land throughout a growing season is known as land use efficiency, and it mostly depends on the length of the crop. The mean LUE under the improved practice was 88.49%, while the mean LUE values were 60.82% and 61.64% for the existing patterns. The use of land by the IP was increased compared to what occurred in both EP1 and EP2 (Table 3). The LUE values for EP1, EP2, and IP in 2017–2018 were 61.24, 61.95, and 88.74%, respectively, while in 2018–2019, those values were 60.39, 61.32, and 88.23%, respectively (Table 4).

**Table 4.** Rice equivalent yield, farmers' land use efficiencies, and production efficiencies of existing and improved cropping patterns in 2017–2018 and 2018–2019.

| Year | Year | Rice Equivalent Yield (t ha$^{-1}$ yr$^{-1}$) | Land Use Efficiency (%) | Production Efficiency (kg ha$^{-1}$ day$^{-1}$) | BCR |
|---|---|---|---|---|---|
| T. *aman*–fallow–*boro* (EP1) | 2017–2018 | 10.35 d | 61.24 b | 42.07 b | 2.17 b |
| | 2018–2019 | 9.39 d | 60.39 b | 40.94 b | 2.21 b |
| T. *aman*–fallow–tomato (EP2) | 2017–2018 | 14.02 c | 61.95 b | 43.90 b | 2.66 a |
| | 2018–2019 | 15.27 b | 61.32 b | 42.81 b | 2.72 a |
| T. *aman*–mustard–mungbean–*aus* (IP) | 2017–2018 | 16.45 a | 88.74 a | 50.76 a | 2.51 a |
| | 2018–2019 | 17.11 a | 88.23 a | 49.45 a | 2.67 a |
| | Level of significance | ** | ** | ** | ** |
| | CV (%) | 5.53 | 2.44 | 5.11 | 3.50 |

In a column, figures with the same letter do not differ significantly whereas figures with dissimilar letters differ significantly. ** = Significant at 1% level of probability.

### 3.4. Production Efficiency

The improved pattern generated significantly greater average production efficiency (PE) (50.11 kg ha$^{-1}$ day$^{-1}$), whereas lower values (41.51 and 43.36 kg ha$^{-1}$ day$^{-1}$ for EP1 and EP2, respectively) were found in both of the existing patterns (Table 3). The increased percentages of PE in the IP over the EPs are shown in Table 3. In 2017–2018, the PE values for EP1, EP2, and IP were 42.07, 43.90, and 50.76 kg ha$^{-1}$ day$^{-1}$, respectively. In 2018–2019, those values decreased to 40.94, 42.81, and 49.45 kg ha$^{-1}$ day$^{-1}$, respectively (Table 4).

### 3.5. Cost Benefit Analysis

Budget and return assessments were carried out in accordance with the current market prices (BDT 85 = USD 1) throughout the crop-growing period. While considering the economical aspect, the improved pattern (IP) exhibited its superiority over the existing patterns (EPs) over the course of both years. The improved cropping pattern had a higher average gross income of BDT 757,820 ha$^{-1}$ compared to BDT 433,800 ha$^{-1}$ for EP1 and BDT 606,000 ha$^{-1}$ for EP2 (Tables 5 and 6). The expenses of cultivation for the IP (BDT 292,685 ha$^{-1}$) were superior to those of the EPs (BDT 198,100 ha$^{-1}$ and BDT 219,600 ha$^{-1}$) because of higher labor wages and greater input costs (Table 4). The gross margin was considerably superior in the IP (BDT 448,715 ha$^{-1}$) compared to in the EPs (BDT 235,700 ha$^{-1}$ and BDT 386,400 ha$^{-1}$) (Table 6). In the case of the marginal benefit cost ratio, the study showed that the IP provided a significantly greater benefit cost ratio (2.59) compared to EP1 (2.19) but a lower benefit cost ratio compared to EP2 (2.69), which was statistically non-significant (Table 3). EP1, EP2, and IP had BCRs of 2.17, 2.66, and 2.51 in 2017–2018 and 2.21, 2.72, and 2.67 in 2018–2019 (Table 4).

### 3.6. Crop Duration

EP1, EP2, and IP required average durations of 222, 225, and 323 days, respectively (excluding the seedling ages of T. *aman*, *boro*, and *aus* rice) to complete the sequence (Table 6). It was found that an IP consisting of mungbean and mustard may be effectively included in a cropping schedule with a 52-day turnaround duration per year.

**Table 5.** Costs of production for T. *aman–fallow–boro*, T. *aman–fallow–tomato*, and T. *aman–mustard–mungbean–aus* cropping patterns.

| Cost Head | Existing Cropping Pattern-1 (EP1) | | Existing Cropping Pattern-2 (EP2) | | Improved Cropping Pattern (IP) | | | |
|---|---|---|---|---|---|---|---|---|
| | Sylheti Pajam | BRRI dhan28 | Sylheti Pajam | HYV Tomato | Binadhan-17 | Binasarisha-9 | Binamoog-8 | Binadhan-19 |
| **Material Cost (BDT ha$^{-1}$)** | | | | | | | | |
| Seed | 1000 | 1000 | 1000 | 8000 | 1000 | 420 | 3600 | 1000 |
| Fertilizer | 9680 | 9120 | 9680 | 12,920 | 5150 | 7185 | 2140 | 7690 |
| Pesticides | 1500 | 1800 | 1500 | 2000 | 1500 | 1000 | 500 | 1000 |
| Irrigation | - | 3000 | - | 3000 | - | 2500 | 1500 | 1500 |
| **Non-material cost (BDT ha$^{-1}$)** | | | | | | | | |
| Labor cost | 64,000 | 68,000 | 64,000 | 80,000 | 64,000 | 40,000 | 32,000 | 64,000 |
| Land preparation | 9500 | 9500 | 9500 | 8000 | 9500 | 8000 | 3000 | 9500 |
| Fixed cost | 10,000 | 10,000 | 10,000 | 10,000 | 10,000 | 5000 | 5000 | 5000 |
| **Total variable cost** | 95,680 | 102,420 | 95,680 | 123,920 | 91,150 | 64,105 | 47,740 | 89,690 |
| **Value of the product (BDT ha$^{-1}$)** | | | | | | | | |
| Grain yield (BDT ha$^{-1}$) | 180,800 | 214,000 | 186,000 | 400,000 | 234,800 | 97,200 | 131,400 | 208,000 |
| Straw yield (BDT ha$^{-1}$) | 19,000 | 20,000 | 20,000 | - | 22,000 | 13,405 | 3015 | 18,000 |
| By product yield (BDT ha$^{-1}$) | - | - | - | - | - | 30,000 | - | - |
| **Gross return (BDT ha$^{-1}$)** | 199,800 | 234,000 | 206,000 | 400,000 | 256,800 | 140,605 | 134,415 | 226,000 |
| **Gross margin (BDT ha$^{-1}$)** | 104,120 | 131,580 | 110,320 | 276,080 | 165,650 | 63,095 | 83,660 | 136,310 |
| BCR | 2.09 | 2.28 | 2.15 | 3.23 | 2.82 | 2.19 | 2.82 | 2.52 |

(BDT 85 = USD 1).

**Table 6.** Average economic performances of T. *aman–fallow–boro*, T. *aman–fallow–tomato*, and T. *aman–mustard–mungbean–aus* cropping patterns.

| Cropping Pattern | Total Variable Cost (BDT ha$^{-1}$) | Gross Return (BDT ha$^{-1}$) | Gross Margin (BDT ha$^{-1}$) |
|---|---|---|---|
| T. *aman–fallow–boro* | 198,100 c | 433,800 c | 235,700 c |
| T. *aman–fallow–tomato* | 219,600 b | 606,000 b | 386,400 b |
| T. *aman–mustard–mungbean–aus* | 292,685 a | 757,820 a | 448,715 a |
| Increased (%) of IP over EPs | EP1 = 47.75 | EP1 = 74.70 | EP1 = 90.38 |
| | EP2 = 33.28 | EP2 = 34.00 | EP2 = 16.13 |
| Level of significance | 2.27 | 2.04 | 3.03 |
| CV (%) | ** | ** | ** |

In a column, figures with the same letter do not differ significantly whereas figures with dissimilar letter differ significantly. ** = Significant at 1% level of probability.

## 4. Discussion

The improved cropping sequence T. *aman–mustard–mungbean–aus* was found to produce greater yields and by-products under improved farming techniques. Both the inclusion of mustard and mungbean as cultivable crops as well as the implementation of modern farming methods for cultivable crops were among the factors that contributed to the increases in yield under the improved farming techniques. These findings were similar to those reported by other researchers [18–21] who showed that the introduction of mustard in a cropping sequence positively influenced cropping intensity. As a result of the addition of mungbean biomass, it has been claimed that the soil's fertility level has improved.

The study by Mondal et al. [22] revealed that a high-yielding mustard cultivar with a short growing season, such as BARI Sarisha14, could be easily cultivated throughout the fallow season. The addition of an improved variety of mustard within a fallow season of the T. *aman–fallow–boro* rice cropping cycle can increase the overall production compared to the present cropping cycle. Again, by discouraging a highly irrigated and time-consuming *boro* rice crop in this cropping pattern, farmers can easily introduce *aus* rice and mungbean crop, which also can improve soil fertility. Generally, puddling, which optimizes rice production, degrades soil structure, raises bulk density, and diminishes hydraulic conductivity, creating a nonconductive soil physical environment for the next crop [23]). In addition, continuous rice farming can also create a hard pan below the plough layer that prevents subsequent crop roots from growing [24]. Fast-growing leguminous crops, like mungbean, can have the

ability to boost and sustain both profitability and productivity in the rice-based cropping system owing to their adaptation to rice-based cropping patterns and their ability to fix nitrogen from the air [24–26]. The addition of green manuring crops is also linked to changes in the soil's bulk density, total pore space, water-stable aggregates, and hydraulic conductivity [24]. Of the component crops, mustard produced the most by-product over the course of an average two-year period. Nazrul et al. [18] noted that the addition of modern cultivars with the latest technology for material farming increased the production of by-products in comparison to the patterns that farmers had previously used.

The REY data revealed that the improved cropping system produced higher REY values than the existing cropping patterns. The overall REY was raised by using the high-yielding cultivar Binadhan-17 in lieu of Sylheti Pajam, introducing two other crops, and employing advanced farming approaches in the improved pattern. Improved cropping cycles yielded greater REY because of the superior market prices of the element crops (mustard and mungbean) under the improved farming practices. The employment of traditional cultivars and farming techniques resulted in the REYs being lower in the two existing patterns [18]. The REYs grew for EP2 and IP from 2017–2018 to 2018–2019, whereas it fell for EP1, but the difference was not significant. Additionally, for EP1 and EP2, the inclusion of mustard and mungbean in IP in the *rabi* season enhanced REY values by 70.01% and 14.58%, respectively (Tables 3 and 4). These results are in agreement with that of Mondal et al. [22], who claimed that adding a third crop (mungbean) to his studied pattern led to a 67% improvement in overall production over farmers' practices.

Higher LUE values were recorded under the improved pattern compared to the existing patterns due to the inclusion of three other crops viz. mustard, mungbean, and *aus* rice. These outcomes were identical to the findings reported by Khatun et al. [27] and Khan et al. [28] due to the inclusion of mustard and garden pea as new crops, respectively. In addition, the LUE was 45.50% higher in the IP compared to EP1 and 43.56% higher in the IP than EP2, mostly because the IP occupid the field for a longer duration (323 days) than the EPs (EP1 = 222 days, EP2 = 225 days) in a year (Table 3). As a result, the IP might make better use of labor than the EPs [28]. The LUE decreased very slightly the following year when compared to the previous year, although the differences were not statistically significant (Table 4).

The greatest production efficiencies (50.31 and 49.91 kg ha$^{-1}$ day$^{-1}$) were documented under the improved pattern in the years 2017–2018 and 2018–2019, respectively (Table 3). Moreover, the improved pattern generated greater average production efficiency (50.71 kg ha$^{-1}$ day$^{-1}$), whereas lower values (34.39 and 35.65 kg ha$^{-1}$ day$^{-1}$) were found in both existing patterns. Nazrul et al. [18] and Khan et al. [28] established that the average production efficiency was, every time, greater in the improved pattern compared to in farmers' practices. Therefore, the improved cropping system's high production efficiency stipulates that when grown using modern cultivation methods, material crops can be left in a field for a long time while still producing large yields. In contrast, minimum production efficiency was documented in the farmers' practices where conventional management was used, indicating that crops with low yields remain in a field for a short period of time under conventional management. In addition, Khan et al. [29] reported that production efficiency in an improved cropping pattern increased by 3.57 kg ha$^{-1}$ day$^{-1}$ over farmers' practices, which might be due to the inclusion of an additional mungbean crop with modern varieties and improved management practices. In addition, when compared to the PEs between years, the LUE marginally decreased the next year, but the differences were not statistically significant (Table 4). In case of BCR, the study showed that IP provided a significant greater benefit cost ratio (2.59) compared to EP1 (2.19), but a lower benefit cost ratio compared to EP2 (2.69), which was statistically non-significant. When it came to EP2, a larger gross return led to a higher BCR. However, compared to the EPs, the gross margin of the IP is significantly greater (Table 6).

The gross margin was superior in the IP compared to EPs. Introduction of mustard and mungbean in the above-mentioned cropping sequences throughout the fallow time,

and by discouraging *boro* rice and tomato cultivations, the gross margin was increased. In case of the marginal benefit cost ratio, the IP achieved a significant higher benefit cost ratio compared to EP1, but a non-significant lower benefit cost ratio compared to EP2. Farmers were really enthusiastic about growing mungbean and mustard. The farmers further stated that the yield performances of Binadhan-17, Binasarisha-9, Binamoog-8, and Bindhan-19 were acceptable. The modern mustard variety (Binasarisha-9) and mungbean variety (Binamoog-8) may be cultivated easily after T. *aman* rice harvest, which does not hinder or delay the cultivation of *aus* rice. Thus, the farmers in the research area may find it to be both economically and agronomically advantageous to include mungbean in their existing patterns. The economic analysis as shown in Table 6 indicated the higher return of the IP than EP1 and EP2. The average gross return of the IP was BDT 757,820 ha$^{-1}$, which as 74.70% and 34.0% higher over the gross returns of EP1 and EP2, respectively. The mean variable cost was lower in EP1 (BDT 198,100 ha$^{-1}$) and EP2 (BDT 219,600 ha$^{-1}$) than that in the IP (BDT 292,685 ha$^{-1}$), which was probably due to the inclusion of mustard and mungbean in the pattern as well as management practices. The average gross margin was substantially higher in the IP (BDT 448,715 ha$^{-1}$) than EP1 (BDT 235,700 ha$^{-1}$) and EP2 (BDT 386,400 ha$^{-1}$). The higher gross margin of the IP was achieved mainly due to the higher yield advantages of the component crops. Additional gross margin (EP1 = 90.38%) and (EP2 = 16.13%) was achieved by adding 47.75% extra cost (for EP1) and 33.28% extra cost (for EP2) in the improved pattern. These findings are supported by Sarker et al. [30] who found that among the six patterns, wheat–mungbean–T. *aman* rice had the greatest economic advantage in terms of BCR. They claimed that because of the presence of mungbean, the improved pattern demonstrated superiority over the farmers' practices. Moreover, due to the presence of mungbean and mustard, IP has a longer crop duration than EP1 and EP2.

## 5. Conclusions

To develop a crop cultivation strategy which is both environmentally and socioeconomically feasible, a deeper understanding of crop growing systems is essential. Utilizing the CP fallow season to boost productivity can sustain agricultural production system improvement. From the above study, T. *aman*–mustard–mungbean–*aus* rice pattern showed better performance than the existing T. *aman*–fallow–*boro* and T. *aman*–fallow–tomato CPs in terms of agronomical and economic performance. Through this effort, the regional distribution of CPs (in the hilly part of Bangladesh) was also discovered. These findings should aid in future planning for CPs that could increase crop productivity in the Khagrachhari district of the Chittagong Hill Tracts. For large-scale production in Bangladesh's hilly regions, the findings might be valuable for academics, outreach workers, and national policy planners. In addition, by growing four crops in a year on the same plot of land, cropping intensity and productivity will increase, leading to more opportunities for both male and female laborers to find employment. At the same time, increased production of rice, mustard, and mungbean will ensure the farmers' and the country's food and nutritional security.

**Author Contributions:** Conceptualization, R.G., M.J.S and M.S.I.; methodology, R.G., M.J.S and I.J.R.; software, R.G., M.J.S and M.S.I.; validation, U.K.S. and M.R.U.; formal analysis, S.M. and F.Z.; investigation, R.G. and M.J.S.; resources, R.G. and M.J.S.; data curation, A.K.H. and A.K.M.M.I.; writing—original draft preparation, R.G., M.S.I. and M.J.S.; writing—review and editing, A.H., G.D.A.-Q., J.A.P. and E.F.A.; visualization, R.G.; supervision, U.K.S. and M.R.U.; project administration, R.G. and M.J.S.; funding acquisition, R.G. and E.F.A. All authors have read and agreed to the published version of the manuscript.

**Funding:** This research was funded by "Bangladesh Institute of Nuclear Agriculture (BINA), grant number: Revenue budget". The authors would like to extend their sincere appreciation to the Researchers Supporting Project Number (RSP2023R134), King Saud University, Riyadh, Saudi Arabia.

**Data Availability Statement:** Not applicable.

**Acknowledgments:** The authors would like to thank Field Man, BINA Sub-station, Khagrachari, SAO, Department of Agricultural Extension, Sadar and Panchhari, Khagrachhari, Bangladesh for their assistance throughout the research. The authors would like to extend their sincere appreciation to the Researchers Supporting Project Number (RSP2023R134), King Saud University, Riyadh, Saudi Arabia.

**Conflicts of Interest:** The authors declare no conflict of interest. The funders had no role in the design of the study; in the collection, analyses, or interpretation of data; in the writing of the manuscript; or in the decision to publish the results.

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
