# Peer review of "Introducing New Cropping Pattern to Increase Cropping Intensity in Hill Tract Area in Bangladesh"

_sustainability, doi:10.3390/su151411471_

Round 1
Reviewer 1 Report
In the manuscript “Introducing New Cropping Pattern to Increase Cropping Intensity in Hill Tracts Area of Bangladesh” authors introduced new cropping system and compared with different production parameters. The article is written well but has some major issues that needs to be considered before accepting for publication:
1. Abbreviations like T. aman should be expanded at first use in abstract as well as in introduction.
2. 2.1. Experimental location and design: Some information like -How many farmers were involved in the experiment? Whether cropping systems were replicated at each farm? Or no. of farmers considered as replication?
3. The data presented in the tables are not statistically compared. Also, there is no information on statistical analysis of data in the methodology section. Then, how the data of different cropping system compared? There is no meaning of presenting data without statistical analysis as you cannot claim the difference whether significant or not.
4. Table 3. Please check the unit of yields
The authors compared only productivity and economics of cropping systems, that gives the article impression as routine work without any novelty. Readers would like to see the effect of different cropping systems on soil fertility/properties, water productivity etc.
-
Author Response
Reviewer 1's observations about the manuscript's shortcomings and comments/suggestions for how to make the document better were much appreciated by the authors. As a result of reviewer suggestions, we revised the manuscript.
Attached File:
Response to Reviewer 1 Comments
Point 1: In the manuscript “Introducing New Cropping Pattern to Increase Cropping Intensity in Hill Tracts Area of Bangladesh” authors introduced new cropping system and compared with different production parameters. The article is written well but has some major issues that needs to be considered before accepting for publication:
Response 1: The authors are grateful to the reviewer 1 for the suggestions to improve the manuscript. In this revison, we have revised our manuscript by following reviewer suggestions.
Point 2: Abbreviations like T. aman should be expanded at first use in abstract as well as in introduction.
Response 2: Done.
Point 3: Experimental location and design: Some information like -How many farmers were involved in the experiment? Whether cropping systems were replicated at each farm? Or no. of farmers considered as replication?
Response 3: Thank you for the suggestions to add those important inforamtions and following reviewer sugestions we have added those infromtions. Three farmers were involved in this experiment. This experiment were conducted in three locations of Khagrachari districts of Bangladesh following Randomized Completely Block Design (RCBD) with three dispersed replication.
Point 4: The data presented in the tables are not statistically compared. Also, there is no information on statistical analysis of data in the methodology section. Then, how the data of different cropping system compared? There is no meaning of presenting data without statistical analysis as you cannot claim the difference whether significant or not.
Response 4: Thank you for the suggestions on statistical analysis. We have performed dtatistical analyis in this revised manuscript.
Point 5: Table 3. Please check the unit of yields
Response 5: We have checked and changed the yield unit in the manuscript.
Point 6: The authors compared only productivity and economics of cropping systems, that gives the article impression as routine work without any novelty. Readers would like to see the effect of different cropping systems on soil fertility/properties, water productivity etc.
Response 6: Thank you very much for giving the suggestions and we have tried our best to improve the manuscript following reviewer suggestions. An detailed study of Bangladesh's three hilly districts, including their productivity and economics of agricultural systems, soil fertility and characteristics, water productivity, and other factors, is being conducted as part of this project. Only cropping patterns are the major focus of this investigation.
Reviewer 2 Report
Thank you, authors, for writing a paper on this topic. The idea, contents and discussions of results of the paper are interesting. However, the paper quality will improve if it is revised. While revising the paper, the author(s) should give attention to few comments and suggestions given below.
(1) The abstract is nice. But it is better to write the specific study objective also in abstract itself.
(2) A brief background information should be given on the cropping patterns of Bangladesh atleast for last one decade. This will help the readers to understand overall situation in Bangladesh.
(3) What are limitations of the present study? Every research study suffers from some limitations. But authors have not mentioned them. Authors should mention some specific limitations like data problem, methods or techniques, a single place case study location etc. preferably in the last section.
(4) What is the theoretical justification of present study? A brief explanation may be given as a footnote or endnote, if not as a section.
(5) Contents in the last section 5 (conclusions) should be more elaborate on policy implications and suggestions. Changing cropping patterns have several consequences (economic, social, etc.) with respect to input-use patterns, requirements of people and output market channels. These changes have issues with policy makers and other stake holders. Such issues should come out.
(6) Experiments and field surveys were done only in one district (Khagrachhari district) with specific soil types and irrigation conditions. How findings and suggestions of this present study will be applicable to other regions with different soil types and irrigation conditions.
(7) Authors should improve the overall analysis and discussions of the results in section 4. Discussions and analysis of results are mainly based on own data. Authors are giving the analysis of results as “it happens” and not giving so much on “why it happens”. Please give more discussions on “why it happens” also.
(8) Some sentences (please see below, for one example) are very long. A long sentence may be split into two or three smaller sentences without losing original meanings.
(Line 87 to 93)
The experiment was executed in two Upazilla of Khagrachhari district in three locations viz. Shalban, Sadar Upazilla, Thakurchara, Sadar Upazilla and North Shantipur, Panchari Upazilla at farmers’ fields where irrigations facilities were available to grow crops and to boost the intensity of cropping, and productivity of the experimental areas with the induced cropping pattern T. aman (Binadhan-17) - mustard (Binasarisha-9) - mung-bean (Binamoog-8) - aus (Binadhan-19) over the previously used cropping pattern T. aman-fallow-boro rice or T. aman- fallow-vegetables during 2017-18 and 2018-19.
----------x-------------
Thank you authors. The paper is well-written.
Author Response
The authors are very much grateful to the reviewer who has critically reviewed our manuscript and following the suggestions and comments given by the reviewer 2, we have tried to revise our manuscript.
Attached File:
Response to Reviewer 2 Comments
Point 1: Thank you, authors, for writing a paper on this topic. The idea, contents and discussions of results of the paper are interesting. However, the paper quality will improve if it is revised. While revising the paper, the author(s) should give attention to few comments and suggestions given below.
Response 1: The authors are grateful to the reviewer 2 for the suggestions to improve the manuscript. In this revison, we have revised our manuscript by following reviewer suggestions.
Point 2: The abstract is nice. But it is better to write the specific study objective also in abstract itself.
Response 2: Thank you for the appreciable comments on abstract. We have added specific study objective in the revised manuscript.
Point 3: A brief background information should be given on the cropping patterns of Bangladesh atleast for last one decade. This will help the readers to understand overall situation in Bangladesh.
Response 3: We have added informations on the cropping patterns of Bangladesh.
Point 4: What are limitations of the present study? Every research study suffers from some limitations. But authors have not mentioned them. Authors should mention some specific limitations like data problem, methods or techniques, a single place case study location etc. preferably in the last section.
Response 4: Thank you for asking this issues, we have added such information in the manuscript as suggested by the reviewer.
Point 5: What is the theoretical justification of present study? A brief explanation may be given as a footnote or endnote, if not as a section.
Response 5: Thank you very much for this suggestions. We have treid to improve the manuscript.
Point 6: Contents in the last section 5 (conclusions) should be more elaborate on policy implications and suggestions. Changing cropping patterns have several consequences (economic, social, etc.) with respect to input-use patterns, requirements of people and output market channels. These changes have issues with policy makers and other stake holders. Such issues should come out.
Response 6: We have revised the conclusion section following reviewer suggestions.
Point 7: Experiments and field surveys were done only in one district (Khagrachhari district) with specific soil types and irrigation conditions. How findings and suggestions of this present study will be applicable to other regions with different soil types and irrigation conditions.
Response 7: Thank you for the nice questions. The Chittagong Hill Tracts in the south-eastern part of Bangladesh comprises a total area of 5,093 sq. miles encompassing three hill districts: Rangamati, Khagrachari and Bandarban. We conducted the expriment Khagrachari fisrt and we have plan to start the same for other twos. Surly, the experince that we got from Khagrachari will help us before going to start in Rangamati and Bandarban. Bangladesh Institute of Neclear Agriculture (BINA) try to disseminate these cropping pattern all the parts of Bangladesh where it is applicable such as medium high land with well irrigation facilities.
Point 8: Authors should improve the overall analysis and discussions of the results in section 4. Discussions and analysis of results are mainly based on own data. Authors are giving the analysis of results as “it happens” and not giving so much on “why it happens”. Please give more discussions on “why it happens” also.
Response 8: Thank you very much, we have tried our best to revised the discussion section followin reviewer suggestions.
Point 9: Some sentences (please see below, for one example) are very long. A long sentence may be split into two or three smaller sentences without losing original meanings.
(Line 87 to 93)
The experiment was executed in two Upazilla of Khagrachhari district in three locations viz. Shalban, Sadar Upazilla, Thakurchara, Sadar Upazilla and North Shantipur, Panchari Upazilla at farmers’ fields where irrigations facilities were available to grow crops and to boost the intensity of cropping, and productivity of the experimental areas with the induced cropping pattern T. aman (Binadhan-17) - mustard (Binasarisha-9) - mung-bean (Binamoog-8) - aus (Binadhan-19) over the previously used cropping pattern T. aman-fallow-boro rice or T. aman- fallow-vegetables during 2017-18 and 2018-19.
Response 9: We have separated such long sentences into single sentences. Thank you for the suggestions here.
Reviewer 3 Report
The foundation for the research is very well explained in the Introduction. The research is very important because it relates to food security – a great concern worldwide today.
Discussions are pertinent, well backed by references, and of importance for agricultural practice.
Some improvements could be made on the research and manuscript:
In line 96: specify the classification system according to which the land type comes under the medium-high category and also add a few words about what that means in terms of fertility.
More and clearer information is needed for the initial soil fertility state. It would be better to give data in Table 1 and comment in the text on the supply classes (low, medium, etc.). Also, specify references for framing the data into supply classes. The soil pH range is very wide, it suggests high heterogeneity of the experimental area. As I understand it, there are three experimental locations so it would be better to present the soil chemical characteristics for each one. Besides, each crop has certain requirements for soil pH so it would be expected that soil reaction greatly influence the experiment. In the same table data and supply classes probably represent intervals for several samples; how many samples? where were they collected from and how? on what depth?
I recommend organizing the experimental fields in replicate plots and applying statistical tools (analysis of variance, for example) in order to demonstrate that the differences are statistically ensured.
Table 2 might be divided by items for more clarity: items referring to fertilization first, next those referring to crop characteristics, and then those reflecting yield.
A social study, based on a Questionnaire, would be required to support the statement regarding farmers’ reaction (subchapter 3.7.). The same goes for the statement in rows 252, 253.
The statement in rows 215, 216 regarding the improvement of soil fertility needs reference or data to back it. As a matter of fact, I was going to suggest extending the research to soil fertility properties changes under the improved cropping pattern.
Author Response
First of all, we would like to thank reviewer 3 for the recommendations and remarks, especially about the soil and farmers' reactions. We made an effort to respond to reviewers' questions and prepare a response letter to comments.
Attached File:
Response to Reviewer 3 Comments
The foundation for the research is very well explained in the Introduction. The research is very important because it relates to food security – a great concern worldwide today.
Discussions are pertinent, well backed by references, and of importance for agricultural practice.
Some improvements could be made on the research and manuscript:
Point 1: In line 96: specify the classification system according to which the land type comes under the medium-high category and also add a few words about what that means in terms of fertility. More and clearer information is needed for the initial soil fertility state. It would be better to give data in Table 1 and comment in the text on the supply classes (low, medium, etc.). Also, specify references for framing the data into supply classes. The soil pH range is very wide, it suggests high heterogeneity of the experimental area. As I understand it, there are three experimental locations so it would be better to present the soil chemical characteristics for each one. Besides, each crop has certain requirements for soil pH so it would be expected that soil reaction greatly influence the experiment. In the same table data and supply classes probably represent intervals for several samples; how many samples? where were they collected from and how? on what depth? I recommend organizing the experimental fields in replicate plots and applying statistical tools (analysis of variance, for example) in order to demonstrate that the differences are statistically ensured.
Response 1: The authors are grateful to the reviewer for the valuable comments on this topic. We did not collect and analyze the soil parameters. We followed maily Fertilizer Recommendation Guide (FRG)-2012 published by Bangladesh Agricultural Research Council, Dhaka. Based on FRG 2012 and available literature on soil properties for experimental area AEZ-29 (Northern and Eastern Hills) we designed our experiment. In the revised version, we have added specifi references.
Point 2: Table 2 might be divided by items for more clarity: items referring to fertilization first, next those referring to crop characteristics, and then those reflecting yield.
Response 2: Done.
Point 3: A social study, based on a Questionnaire, would be required to support the statement regarding farmers’ reaction (subchapter 3.7.). The same goes for the statement in rows 252, 253.
Response 3: We appreciate your interest in this topic. Three farmers' fields were used for the experiment. They were really pleased to see the improved cropping pattern's yield performance, and we have included the feedback here. If the reviewer suggests that this be removed, we would like to remove the farmers' reactions in our revised manuscript later.
Point 4: The statement in rows 215, 216 regarding the improvement of soil fertility needs reference or data to back it. As a matter of fact, I was going to suggest extending the research to soil fertility properties changes under the improved cropping pattern.
Response 4: I appreciate your advice in this case. As the current experiment was done a few years ago and for several limitations, we amended the discussion but could not to undertake a soil test again. However, we greatly valued the insightful criticism from reviewers.
Reviewer 4 Report
The data obtained by the authors may be good, which include biophysical and economical data, but to be a good publishable scientific paper, the overall methods need to be revised followed by analyzing the data using a relevant statistical analysis method, such as ANOVA (Analysis of Variance) and a relevant means test. Then please develop the discussion of the results based on the statistics.

Minor language editing is required.
Author Response
First of all, thank you so much for the suggestion to analyze some manuscript data. We made an effort to write the manuscript after analyzing the crops' grain and straw yields as well as other factors that the reviewers recommended. We used the present market value and the BCR calculation techniques, and we only performed analysis on yield-related metrics, not product values.
We apologize for not writing the entire reviewer's comments or words because they are not editable; instead, we merely highlighted the LINE number in the response letter.
Attached File:
Response to Reviewer 4 Comments
Point 1: Line 87-88
Response 1: The reviewers' insightful criticism and recommendations for how to make the manuscript better were much appreciated by the auhtors. Following reviewr suggestions, we have clarified the experimental design and layout used in the study.
Point 2: Line 163
Response 2: Again, we are really appriciated the reviewer for his critical observations on the statiscal analysis of the manuscript. Though, initially we followed some literature on how to present the data (where no statistical analysis was done), but now in the revised manuscript we have presentaed data after making data analysis.
Point 3: Line 171, 173-175, 188, 201
Response 3: We have analyzed the data and perfomed ANOVA, and cleared the replication were done for each cropping pattern.We have made other changes also as suggested by the reviewer.
Point 4: Line 208, 210, 217, 227, 232, 238, 245, 258-259,
Response 4: Thank you for pointing this. We have modified the references as suggested by the reviewer and auhtor instructions and re-arraged the list of references.
Point 5: Line 233-235, 236-238
Response 5: Thank you very much for the suggestions here and we have revised our manuscript folloing reviwer suggestions.
Point 6: Line 264
Response 6: Thank you very much for giving the suggestions on conclusions. We have rewritten the conclusion.
Round 2
Reviewer 1 Report
I appreciate the efforts of the authors to address and accepting the changes.
However, I could not find the additional data/information on water productivity etc. as authors replied to have incorporated these data in manuscript.
Kindly recheck.
English Language is fine accept few errors.
Author Response
The authors are grateful to the reviewer for the comments and suggestions and following reviewer suggestions we have improved the manuscript. Responses from the authors are included in the attachment.
Attached File:
Response to Reviewer 1 Comments
Point 1: Point: I appreciate the efforts of the authors to address and accepting the changes.
However, I could not find the additional data/information on water productivity etc. as authors replied to have incorporated these data in manuscript.
Response 1: The authors are grateful to the review 1 for the valuable comments and suggestions to improves the manuscript. In our revised manuscript we have added cropping pattern data not water prodcutivity. In our research, we focused manily cropping pattern in the hilly area. We did not measure the crop yield against water consumption. However, we would like to thank the reviewer for the advice, and we will consider reviewer suggestions in our upcoming research projects in hilly areas.
Reviewer 3 Report
I still think it would be better to give data in Table 1 and comment in the text on the supply classes (low, medium, etc.) and also to specify references for framing the data into supply classes.
Explain how was farmer’s reaction (subchapter 3.7.) registered.
English Language quality is good.
Author Response
The authors are grateful to the reviewer for the comments and suggestions and following reviewer suggestions we have improved the manuscript. Responses from the authors are included in the attachment.
Attached File:
Response to Reviewer 3 Comments
Point 1: I still think it would be better to give data in Table 1 and comment in the text on the supply classes (low, medium, etc.) and also to specify references for framing the data into supply classes.
Response 1: Thank you again for the suggestions on the manuscript. Following the reviewer 3 suggestions, we have added data of elements and comment in the text on the supply classes following Fertilizer Recommendation Guide (FRG)-2012) published by Bangladesh Agricultural Research Council, Dhaka for experimental area AEZ-29 (Northern and Eastern Hills). We are sorry as we didn’t clearly understand it previously.
Point 2: Explain how was farmer’s reaction (subchapter 3.7.) registered.
Point 2: We promised to remove the 3.7 sub-shapter if the reviewer suggested it to remove in our previous response. As a result, we removed this subsection from this revision, as the reviewer 3 advised.
Reviewer 4 Report
Dear authors,
Although in your response to reviewer's comments said that the data have been reanalyzed, in fact relevant data analysis shown in the revised manuscript does not proof it. The main conclusion has not been clearly supported by results of data analysis. BCR for example has to be analyzed also using ANOVA and relevant means test to be able to properly draw the conclusion.
Please find the reviewer's comments on the file attached, and please also check back those previous comments, which are not included in this file.

The quality of language of this paper will have to be revised, especially for the wrong words, wrong structure of the sentences, as well as paragraphs.
Author Response
The authors are grateful to the reviewer for the comments and suggestions and following reviewer suggestions we have improved the manuscript. Responses from the authors are included in the attachment.
Attached File:
Response to Reviewer 4 Comments
Point 1: General comment:
This revised version of the manuscript still does not show a clear easy-to-read article. A good paper should clearly show the objective(s) of the study by explicitly concluding the Introduction with the statement of the objective(s), and based on the objective(s), the authors will be able to clearly present the design of the study. These will make it easy to analyze the data, to present the analysis results and interpretation of them in the discussion. These all are needed to present relevant conclusion and the implication of the study. So, this paper still needs further revisions, as explained in the following comments. Please also use appropriate words in writing sentences; clear and meaningful statements; and good structure of paragraphs and logical link between paragraphs (Not too long paragraph, as previous comments). Here are some examples. The sentence in line 46-48: Consequently, to determine the optimal use of the accessible resources for maximizing the net benefits subjected to some constraints [2]. This is not a complete sentence, so it’s not clear what the authors trying to say. It’s also not clear that this sentence is a consequence of the previous sentence. The sentence in line 48: Crop optimization has established widespread attention... The word “has established” is not the appropriate words; how can “Crop optimization” establishes something? The sentence in line 49-50: The key factors for the sustainable agricultural development of a nation are land and water [1]. This is too vague statement. The “sustainable agricultural development” is too wide term, and the key factors cannot be “land and water”; it must be some aspects of land and water. In addition, this paper also does not discuss about land and water. There are also many typing errors, please fix them.
Response 1: Once more, the authors are appreciative of the reviewers' constructive critique and suggestions for how to improve the manuscript more.. First we are sorry as we didn’t clearly the reviewer suggestions and comments on the manuscript specially on research design, analysis (particularly on BCR and its subsequesnt discussion). We attempted to address all of these in the present revised manuscript and made changes in response to the comments and recommendations of the reviewers 4. We have rewritten the sentences in response to the reviewer's general comments and suggestion. We have removed any words that are either irrelevant or difficult to understand.
Point 2: Line 70-115, This is too long paragraph; it has to be separated into two or three relevant paragraphs.
Response 2: It's been done.
Point 3: Line 133-135, Comments: An experiment has to have at least three replications. In this paragraph, please clarify and state clearly how many replications were made for each cropping patterns, and please explicitly state which location was replication #1 for all cropping patterns, which location was replication #2, etc. The results of data analysis should be able to separate whether the cropping pattern effects and the location effects were significant. A scientific paper must be based on conclusions of analyses with a minimum confidence interval of 95%, so the conclusion must be supported with relevant statistical analyses.
In the two lines, … in three locations viz. Shalban, Sadar Upazilla, Thakurchara, Sadar Upazilla and North Shantipur...(line 134-135). Are they three? Please make sure that the readers will understand that the experiment was in three locations, which was the location 1, location 3 and location 3 (I cannot read them as three locations). I think there are two factors tested in this experiment, namely cropping patterns (consisting of three levels) and years of cropping consisting of two levels, namely 2017-18 and 2018-19. And this can be used to evaluate which cropping pattern (CP) was the most sustainable in the longer time period. However, the replications are not clear. Can the three locations become the three replications? Please look at the data. To be able to analyze the data with ANOVA, there have to be the same number of replications for each treatment combination. If locations of the practice of the three CP in each year was the replications, then the authors are able to analyze all data using a factorial ANOVA, which will show the main effects of CP, main effects of year and the interaction effects, as I commented on the previous peer review report.
Response 3.2: We apologize for not addressing all of these details in our amended manuscript. Now we have added those inforamtion clealry. The experimnet was one factorial (cropping pattern) and replicated in three locations of two Upazilla (Sadar Upazilla and Panchari Upazilla). We conducted same experiment next year following. In each location (both year), cropping pattern was replicated three times. During analyis, we followed one way single factor ANOVA. We average two year data and didn’t check the effects of year on cropping patterns.
Point 4: Line 209-211, MSTAT-C, Excel, DMRT ….
Response 4: Thank you for pointing this. We performed one-way single factor ANOVA using Statistix 10, and Tukey’s to measure variance. LSD performed to compare means. Inadvertently, we wrote DMRT.
Point 5: Line 224, Table 2. Grain yield performance of different cropping pattern at 2017-18 and 2018-19)
Line 231, Table 4. Average agronomic performance...
Response 5: Thank you very much for the suggestions here and we have revised our manuscript following reviewer suggestions. We have deleted Table 2, 3 and 4. We have made a new Table 2 comprosing average yield data, by-product data, duration and turnaround days as we discussed on these data in the manuscript. We have put BCR data in Table 3 along with REY, LUE and PE and we performed anlysis all these data incluidng BCR.
Point 6: Line 238, Table 5. Rice-equivalent yield, land use efficiency…
Response 6: Thank you very much for giving the suggestions on conclusions. We have incorporated Table 5 with BCR. The REY, LUE, PE and BCR was statistically analyzed and we discussed based on analysis. BCR of IP was lower than that of EP2, but it was statistically non-significant. But, BCR of IP was statistically sifnificant than that of EP1.
Point 7: Line 360-374 Conclusions
Response 6: Since we have already analyzed the pertinent data, including the BCR, our conclusion appears to be valid in light of the BCR analysis and others. The author would like to thank reviewer 4 for their insightful suggestions and criticism that helped to improve the paper.
Round 3
Reviewer 4 Report
This revision in terms of the language is much better and the English is already good. The readability has improved significantly.
Unfortunately data analysis is getting worse. By analyzing the data averages between the two year experiment really remove the eligibility for this paper for publication in the journal "Sustainability" because of no analysis and discussion related to sustainability.
Averaging data between years is not a good technique of data analysis for an article that is going to be published in the journal “Sustainability”. To be relevant or acceptable for publication in the “Sustainability”, the paper should be able to statistically show which CP is the best in improving the sustainability of crop production, i.e. with productivity are stable or improved between years.
Why the authors spent a lot of time and cost for conducting this experiment for two years but at the end only show the average between year. The average between years scientifically mean nothing. However, if from the two-year experiments, the authors can statistically show the productivity (for example) of a CP was stable or increasing with years, then the paper would be suitable to be published in the “Sustainability”.
To improve the eligibility for publishing this paper in the "Sustainability" please read and consider the reviewer's comments in the attached file.

Author Response
We appreciate reviewer 4's recommendations for improving our work. Thank you very much for continuous suggestions.
Attached File:
Response to Reviewer 4 Comments
Point 1: General comment:
Please provide an approximate range of the month in a year for each of those seasons, because if published, this article will be read by readers from over the world, not just by those from Bangladesh and India. Author of an international journal should keep this principle in mind for readability and citation of their article and the journal.
Response 1: Thank you for pointing this. We have added information regarding the month of those seasons.
Point 2: What does it mean??? The experimental areas are located in the Northern and Eastern Hills
(AEZ-29) of Bangladesh?
Response 2: Again, many thanks. Regarding this, a correction has been made.
Point 3: What is the altitude? How many%? High, medium, or low? In which area/location?
Response 3.2: We have added above information as suggested by the reviewer. For location, we have specified the sites.
Point 4: This mean that gross income, gross margin, total variable cost, and BCR were recorded in each crop for each treatment of CP and each replication. Therefore, all of these data can be statistically analyzed using Analysis of Variance and mean test using Tukey’s HSD. To be confidence in drawing the conclusion in a scientific paper, the decision must be based on the results of statistical analyses, in this case based on ANOVA and Tukey’s test or LSD, because value differences may not be statistically significant, and if so, the authors cannot conclude, for example, CP#3 resulted in the highest gross margin, before proving if they are statistically significant different. So, why the results in Table 5 were not based on statistical analyses (ANOVA & Tukey’s test)???
Response 4: We have analyzed the data of Table 5 (now in Table 6).
Point 5: Actually the use of LSD is not needed if Tukey’s test has been done. Only one of them is needed (Tukey’s test or LSD).
Response 5: Thank you very much for the suggestions here and we have revised our manuscript following reviewer suggestions.
Point 6: Averaging data between year is not a good technique of data analysis for an article that is going to be published in the journal “Sustainability”. To be relevant or acceptable for publication in the “Sustainability”, the paper should be able to statistically show which CP is the best in improving the sustainability of crop production, i.e. with productivity are stable or improved between years. Why the authors spent a lot of time and cost for conducting this experiment for two years but at the
end only show the average between year. The average between years scientifically mean nothing. However, if from the two year experiments, the authors can statistically show the productivity (for example) of a CP was stable or increasing with years, then the paper would be suitable to be published in the “Sustainability”.
Response 6: Thank you very much for giving the suggestions. It will not be justifiable for this kind of experiment as the components of cropping pattern were different (For existing pattern (EP) its two, but improved pattern (IP) it was four). But We compared the REY, LUE, PE and BCR between years and presented in Table 4. In Table 3, we have presented the increased percentage of improved pattern over the EPs. We believe now its clear to understand, how was the productivity of IP over the EPs. Additionally, it is currently accepted that REY, LUE, PE, and BCR are sustainable in both the first and second years.
Point 7: The longer land use will automatically increase cost and level of exploitation of the soil, and this is not good for long term cropping, unless it can be proven land productivity and BCR stable or
increased in the following year. The BCR was also relatively higher in the CP#2 than CP#3.
Response 7: We may now infer that the aforementioned parameters are stable in the initial year and the succeeding year after analyzing REY, LUE, PE, and BCR in all subsequent years Table 5, 6). Although BCR of IP is smaller than EP2, it was determined to be non-significant, and we also discussed it in the manuscript.
Point 8: The cost for using the land has not been calculated; more frequent cropping will increase more cost to be spent for using the land.
Response 8: Since the farmer's own field was employed as the research site, we didn't include in the overhead cost of land. Instead, we considered the material input costs. However, we will take the reviewer's advice into account for our subsequent studies.
Point 9: Gross margin of CP#3 is higher than CP#2, but was this significant? Did it increase with years?
The cost of CP#3 is also higher that CP#2; will the farmers ready to spend more cost before harvest of the yield that normally declines in its price at harvest?
Response 9: It was found to be significant after our analysis. We did not assess the gross return and margin year by year but rather set the same price for both years because they depend on the current market pricing. As IP demonstrates a significant gross margin compared to EPs, we believe that farmers are willing to incur additional expenses in order to increase their revenues. In table 6, we clarified it further along with discussion.